# Vaccine Design against Chagas Disease Focused on the Use of Nucleic Acids

**DOI:** 10.3390/vaccines10040587

**Published:** 2022-04-12

**Authors:** Edio Maldonado, Sebastian Morales-Pison, Fabiola Urbina, Aldo Solari

**Affiliations:** 1Programa de Biología Celular y Molecular, Instituto de Ciencias Biomédicas, Facultad de Medicina, Universidad de Chile, Santiago 8380453, Chile; fabi.urbina1516@gmail.com; 2Centro de Oncología de Precisión (COP), Universidad Mayor, Santiago 7560908, Chile; seba.morales.p@gmail.com

**Keywords:** DNA vaccine, RNA vaccine, vaccine delivery, immune responses, pandemic

## Abstract

Chagas disease is caused by the protozoan *Trypanosoma cruzi* and is endemic to Central and South America. However, it has spread around the world and affects several million people. Treatment with currently available drugs cause several side effects and require long treatment times to eliminate the parasite, however, this does not improve the chronic effects of the disease such as cardiomyopathy. A therapeutic vaccine for Chagas disease may be able to prevent the disease and improve the chronic effects such as cardiomyopathy. This vaccine would be beneficial for both infected people and those which are at risk in endemic and non-endemic areas. In this article, we will review the surface antigens of *T. cruzi*, in order to choose those that are most antigenic and least variable, to design effective vaccines against the etiological agent of Chagas disease. Also, we discuss aspects of the design of nucleic acid-based vaccines, which have been developed and proven to be effective against the SARS-CoV-2 virus. The role of co-adjuvants and delivery carriers is also discussed. We present an example of a chimeric trivalent vaccine, based on experimental work, which can be used to design a vaccine against Chagas disease.

## 1. Introduction

Chagas disease was first described by the Brazilian researcher Carlos Chagas at the beginning of the 20th century (1908) and is endemic to Latin America, although human migration has resulted in the appearance of the disease in North America, Western Pacific areas, and Europe [1]. It is caused by a parasitic protozoan flagellate *Trypanosoma cruzi* and transmitted by the hematophagous triatominae insects, usually *Triatoma infestans* [1]. Worldwide, it is estimated that 8–10 million people are infected by *T. cruzi*, including 300,000 in the United States and 100,000 in Europe [2].

It is also, estimated that 120 million people are at risk of infection from living in endemic areas. The disease has two phases, which are the acute and chronic phases. Once the acute phase resolves, about 30–40% of patients can develop a chronic stage of the disease, in which 30% of them can progress to a severe cardiomyopathy, though mega viscera and polyneuropathy can also develop [3]. Despite a century of research on this deadly disease, many issues still remain unresolved, because diagnostic and epidemiological control, prognostic methods and therapeutic treatments are far from ideal. Moreover, nowadays the population is older and has comorbidities such as immunodeficiencies and cancer or other pathologies that can influence the disease outcome.

Anti-trypanosomal drugs are used to treat acute and congenital Chagas disease, reactivated infections, and chronic disease in children under 18 years old [1,2,3]. The two drugs that are used to treat Chagas disease, are benznidazole (BZN) and nifurtimox (NIF), and they have been used for almost half a century, however, their safety and efficacy profile are not completely ideal. Moreover, their efficacy during the chronic phase is limited. NIF is no longer recommended to treat Chagas disease and BZN is preferred because of its better tolerability profile, tissue diffusion and efficacy [1,2,3]. Nowadays, several drugs are being developed to treat Chagas disease, and a promising series are those named target-based drugs [4] and those based in azoles, nitroimidazoles, oxaboroles and protease inhibitors [5].

Vaccines are biological preparations, which can provide active adaptive immunity against infectious diseases and are one the most effective methods to prevent infectious disease. Vaccination is largely responsible for the worldwide eradication of infectious diseases such as smallpox and the restriction of others such as polio, tetanus, rabies, measles and several others around the world. It has been reported that the World Health Organization (WHO) has licensed vaccines against several preventable infectious diseases [6]. There is growing recognition that a complete interruption of transmission of *T. cruzi* to human is not an easy goal to achieve, and modeling studies suggest that vector control strategies should be combined with other efforts to improve access to better health care for patients. This is in order to reach the goals of the WHO 2020 London declaration, that called for a 100% certified interruption or complete control of Chagas disease [7]. Despite the success of vaccines against several infectious diseases, there are no available vaccines for Chagas disease mainly due to the weak immune response of the host against *T. cruzi* and the several strategies that the parasite has developed to escape the host immune system. In this review we will present recent strategies to develop nucleic acid-based vaccines encoding antigen candidates to obtain a vaccine against Chagas disease in an effort to restrict the parasite spreading and to prevent the clinical outcome of the disease.

## 2. Vaccine Rationale

A therapeutic vaccine would represent an attractive opportunity to improve the care of chagasic patients or to prevent the disease [8]. There are several comparative advantages with the available treatments that exist such as, reduction of toxicity in patients, higher efficacy to prevent cardiac and gastrointestinal complications, prophylactic prevention of Chagas disease and potential use during pregnancy to prevent congenital Chagas disease. An economic analysis of the development of a therapeutic vaccine showed that it is highly cost-effective, would save lives and costs under a wide range of efficacy conditions that delay Chagas disease clinical outcomes [9,10]. Recent opinions regarding the development of a vaccine against Chagas disease and vaccine production can be found in Camargo et al. [11].

A *T. cruzi* vaccine candidate molecule should have at least the following characteristics: (i) to be highly immunogenic, (ii) it has to be an essential molecule for the etiologic agent and contribute as a molecular target to elicit neutralizing antibodies, (iii) it has to be expressed in all parasite stages existing in the vertebrate host (amastigotes and blood trypomastigotes), (iv) the immunogenic molecule should be located at the parasite surface, as long as possible and (v) the candidate molecule should not undergo mutations. Here, in this section we review information on aspects about subunit vaccines and *T. cruzi* antigens that have been used to design vaccines during the last years.

### 2.1. Trypanosoma Cruzi Surface Antigens

*T. cruzi* membrane proteins have been shown to play an important role in *T. cruzi* biology, including the interaction between the parasite and the vertebrate host necessary for parasite infection, survival and proliferation. However, many of them have been described as immunogenic and virulence factors, which were identified by immunological screening of cDNA expression libraries using immune sera from chagasic patients [12,13]. One of the most widely distributed antigens on the parasite surface are the mucin family of proteins, which are useful for sero diagnosis [14,15]. *T. cruzi* is covered by a dense layer of mucin-type molecules which are glycoproteins and their sugar moieties are able to interact with mammalian cells, and they are distributed over the complete parasite surface in the different developmental stages [15]. The most important are those mucins which play a key role in parasite protection as well as in infectivity and modulation of the host immune response [14,15]. *T. cruzi* mucins can be divided into two types, named TcMUC and TcSMUG. The TcMUC can be divided into three groups (I-III) according to their central domains. TcMUC I and II proteins are distributed and present in amastigotes and blood trypomastigote forms and TcMUC I is one of the main components of the amastigote form, while TcMUC II is predominantly present in membrane lipid rafts of the trypomastigote form [15]. TcMUC I proteins contain internal tandem repeats in their structure with a T_8_KP_2_ amino acid sequences, which are targets for the O-glycosylation pathway in *T. cruzi* [15]. The repeated sequence is flanked by an N-terminal signal peptide and a C-terminal glycophosphatidylinositol (GPI)-anchor signal [16,17]. Meanwhile, TcMUC II proteins share similar N and C-terminal regions with TcMUC I, but they lack the internal repeated motif T_8_KP_2_, although they do possess regions with T, K and P which are rich in those amino acid residues [17]. On the other hand, the single gene product of the TcMUC III group, is named trypomastigotes small surface antigen (TSSA) and has been identified as a mucin-like glycoprotein, 20 kDa in size, which is present in mammalian-derived stages of the parasite [15,17]. The second mucin family member TcSMUG, contains a putative signal peptide at the N-terminus and a GPI-anchor signal at the C-terminus, and can be divided into two groups, a small (S) group and large (L) group according to their size [18,19,20]. The S group is composed of 35–50 kDa N-glycosylated mucins (Gp35/50 mucins), which are the main acceptors of sialic acid from *T. cruzi* trans-sialidases on the parasite surface. The S group is found in epimastigotes and metacyclic trypomastigotes, while the TcMUG L group is comprised of non-sialic acid acceptors which are only present in epimastigotes, on the cell surface [21,22].

The other large superfamily of surface antigens are the trans-sialidase (TS) gene products, which contain at least 1.430 gene members, including 693 pseudogenes [23,24,25]. Comparable to mucins, TSs are distributed along the cell body flagellum and flagellar pocket of the parasite. TSs are part of one of the largest super gene families in *T. cruzi*, with hundreds of genes expressed at the same time on the parasite surface that may generate a smoke screen effect to the immune response with the highly dominant TS epitope with the consensus sequence DS_2_AH(S/G)TPSTP(A/V) [26,27]. The TS activity involves the transfer of sialic acid from the host glycoproteins, mainly to the parasite mucins of the trypomastigote cell surface; meanwhile, neuraminidase (TCNA) activity is detected when non suitable acceptor molecules for sialic acid are present, and sialic acid is transferred to water [28]. The sialylation process in *T. cruzi* is crucial for its viability and proliferation in the host [29]. Moreover, it is thought that TS allows assimilation of them and masks them as a mammalian protein to avoid recognition and parasite lysis, and can actively participate in host cell invasion [29,30,31].

The *T. cruzi* TS superfamily can be divided into four groups (I-IV) according to their characteristic motifs (Figure 1). Group I comprises proteins with TS and/or TCNA activity and comprises the TCNA (neuraminidase), SAPA (shed acute-phase antigen) and TS-epi (Figure 1) [17,32]. They are anchored by GPI-anchors to the parasite plasma membrane [17]. TSs can be found in serum from infected mammals, but are also released into the extracellular space via extracellular vesicles [30,33]. When SAPA is released after cleavage of the GPI anchor, similar to other surface antigens, it is speculated that the protein sheds the TS epitope, which is made up of 14 tandemly repeated residues at the C-terminus of SAPA (see Figure 1). That antigen can act as a diversion for the immune system to concentrate the antibody response against it, however, the enzymatic activity is preserved [31,34,35,36]. This speculation has been called the “smoke screen effect” of the parasite to evade the host immune response and is one of the main reasons by which TSs might not be the best target for immune interventions [31]. In early studies on infections of cultured cells by *T. cruzi* trypomastigotes but not epimastigotes, it was observed that high density lipoproteins (HDL) from the culture media, or when supplemented into the media, enhanced infection in a dose-dependent manner [37]. From previous studies it was determined that TcNA was inhibited by the main protein component of HDL, the apolipoprotein A-I [38,39]. Thus, it was hypothesized that the TcNA activity modulates infection through a negative control mechanism, where release of sialic acid by the endogenous neuraminidase would down-regulate infection, and addition of sialic acid to glycoconjugate acceptor(s) should have the opposite effect (up-regulation). TS-epi, a third member of Group I, is an active TS expressed in the insect dwelling epimastigote form and it is different from the TS expressed on the blood trypomastigotes. TS-epi lacks SAPA repeats and is not anchored to the membrane by GPI and it is predicted that anchoring to the membrane is due to the presence of a transmembrane domain followed by a hydrophilic section in the C-terminus [17,40]. The last feature may explain why TS-epi is minimally secreted into the medium [17,40].

The TS group II comprise members of the GP85 surface glycoproteins, including ASP1, ASP2, TSA1, Tc85, SA85, GP82 and GP90, all of which have been implicated in host-cell attachment and invasion. These proteins share a common Asp motif (SxDxGxTW), and the VTVxNVxLYNR motif, which is characteristic of all TS members (see Figure 1). ASP1 and ASP2 are two amastigote surface proteins and TSA1 is a trypomastigote surface protein. Those proteins, ASP1, ASP2 and TSA1 induce strong antibody responses and they are targets of *T. cruzi*-specific CD8^+^ cytotoxic T lymphocytes [17]. The Tc85 glycoprotein molecule (85 kDa) is present in blood trypomastigotes and has been identified as a ligand with the ability to bind to different host receptor molecules (laminin, fibronectin and cytokeratin) located on the cell surface of monocytes, neutrophils, and fibroblasts [17,41,42,43,44]. Meanwhile, SA85 is expressed in amastigotes and blood trypomastigotes, however only the amastigote form can express the mannose-binding protein ligand, which is probably involved in the opsonization of the parasite to enhance its infection capacity [45]. Finally, GP82 and GP90 are glycoproteins, which are expressed mainly at the plasma membrane of metacyclic trypomastigotes, a parasite form excreted by the insect vector [46]. However, GP90 is also present in mammalian blood trypomastigote and amastigote forms and has an antiphagocytic effect mediated by the removal of sugar residues necessary for *T. cruzi* internalization. The mechanism of action seems to be due to a glycosidase activity of GP90, which down-regulates host-cell invasion in a receptor-mediated manner [17].

The TS III group is composed of several surface glycoproteins present in mammalian trypomastigotes (CRP, CEA, TESA, and FL160, Figure 1) and they are recognized by sera from chagasic patients. They are all able to inhibit the classical and alternative pathway of complement activation, which could be a protection mechanism from host complement lysis of the parasite trypomastigote form [47,48]. TESA (trypomastigote excretory-secretory antigen) is located on the cell surface membrane of trypomastigotes, while the other three proteins are flagellum-associated membrane proteins [49,50]. The TS IV group is composed of genes that encode trypomastigote surface antigens, whose function is unknown as yet. This group is included within the TS family since it has the VTVxNVxLYNR, a signature motif of all known TS proteins (Figure 1). The B5 peptide from the Tc13 protein is highly immunogenic and is present in the metacyclic trypomastigote form [26].

The TcTASV (trypomastigote alanine serine valine-rich) proteins are a family that comprises at least 40 members in *T. cruzi* [51,52,53,54]. They all share conserved N and C-terminal domains with a variable central core domain which is rich in Ala, Ser and Val residues, however this domain contains a conserved Ala-Glu-Pro motif. It also has a high number of Ser and Thr susceptible to glycosylation and a signal sequence for the addition of a GPI anchor. It is believed that they are on surfaces located in the parasite, or they can be secreted into the medium. Orthologues for these proteins have not been found in other trypanosomatids. The TcTASV can be divide into four subfamilies (A–D). The TcTASV-C sub family is expressed in trypomastigotes, is phosphorylated and glycosylated with a size of 60 kDa and is attached to the parasite surface (cell body and flagellum) by a GPI anchor, which may explain why it is shed into the medium and in contact with the immune system of the host [17,51,52,53,54]. The TcTASV-A subfamily has been demonstrated in blood trypomastigotes, although a conserved peptide has been found in amastigote and trypomastigote extracts [17,51,52,53,54].

Other interesting antigens are encoded by low copy number genes, such as Tc52 a highly conserved protein, which has glutathione S-transferase activity with immunomodulatory properties and it is essential for cell viability, as the knockout of both alleles is lethal for the parasite [31,54,55,56,57]. This antigen is expressed in all developmental forms of the parasite; however, the highest expression levels are found in the replicative forms of *T. cruzi* (epimastigotes and amastigotes) [31,54]. Immunization based on Tc52 conferred protection against *T. cruzi* challenges [31,58,59,60]. Anti-Tc52 antibodies are able to generate trypomastigote lysis through complement activation, which are neutralizing antibodies and an ideal anti-*T. cruzi* antigen candidate [31]. Immunization with the Tc52 N-terminal domain conferred greater protection than the C-terminal domain or full-length protein in the acute and chronic phases of infection [31,60].

Another very interesting antigen of *T. cruzi* amastigotes and trypomastigotes is Tc80, a prolyl oligopeptidase, which is expressed in extracellular blood trypomastigotes and the intracellular amastigote forms [31]. This enzyme is able to degrade components of the extracellular matrix (collagen and fibronectin) of the host cells which may contribute to host cell invasion [61]. Specific inhibitors of Tc80 were able to block the *T. cruzi* infection in host cells [62]. Tc80, despite being of low immunogenicity in natural infections, immunization with an adjuvant enabled Tc80 to elicit a strong humoral and cell mediated immune response. The humoral response generated antibodies with enzyme inhibition properties, neutralization of *T. cruzi* infectivity in vitro, and complement-mediated lysis of trypomastigotes [63]. Tc80-based vaccines reduced parasite load in acute and chronic phases of Chagas disease, increased survival of mice, and prevented chronic phase-related complications [31,63].

Cruzipains (GP57/51, Cz), the main cysteine proteases of *T. cruzi*, display several optimal criteria for a vaccine candidate and generate a strong immune response in humans [17,31,64]. There are approximately 100 copies of these genes that encode several Cz isoforms, which are highly similar at the amino acid sequence level [17,64]. Specific irreversible enzyme inhibitors of Cz were evaluated in trypomastigote-infected heart muscle cells and proved to interfere with cell invasion and inhibit intracellular *T. cruzi* cell proliferation [65,66]. Cz isoforms are expressed on the body surface of epimastigotes and trypomastigotes, however, Czs are only present in the flagellar pocket and within the pocket regions of trypomastigotes [17,67]. Czs are also found inside extracellular vesicles where they are together with 265–345 other secreted/excreted proteins [33]. The N-terminal portion of Cz has the ability to cleave cysteines present in the Fc portion of immunoglobulins, and therefore can potentially have an immune escape mechanism to avoid complement fixation and antibody-dependent cytoxicity [68]. The N-terminal portion of Cz elicits a specific immune response and confers better levels of protection in vaccinated mice against a lethal challenge of *T. cruzi* than the immunodominant C-terminal portion of the recombinant protein, which mainly distracts host antibody responses [69,70]. Interestingly, Cz-based vaccines can reduce parasitemia and provide high survival rates and most importantly, prevent chronic phase-related damage [31,71,72,73]. Czs have an important role in the parasite’s process of host cell internalization, and inside the macrophage participates in the parasite’s escape from the phagosome to the cytoplasm where the infective trypomastigotes differentiate to amastigotes and can proliferate [31,71]. It is thought that Cz, together with other parasite secreted proteins, represent virulence and immunostimulatory factors to be delivered into host cells and are frequently found by immunological screening of cDNA expression libraries [12].

Additional surface antigens are those belonging to the TcGP63 family and amastin family. The TcGP63 family comprises members with zinc-dependent metalloproteinase activities and has at least two groups of proteins (I and II). The TcGP63-I group of proteins is present in all three developmental stages of *T. cruzi* and demonstrate metalloproteinase activity [74]. They are bound to the parasite membrane by means of a C-terminal GPI anchor. Two isoforms of the TcGP63-I are known, which are a glycosylated form and a non-glycosylated form. The glycosylated form is present on the surface membranes of epimastigotes and amastigotes and is also irregularly present on the surface of flagellum and cell body membranes of epimastigotes [17,74]. The non-glycosylated form is intracellular and present near the kinetoplast and in the flagellar pocket of metacyclic trypomastigotes. The amastin family of proteins is a transmembrane group of small proteins, which has four subfamilies (α, β, γ and δ). The exact biological role of amastins is still unknown, although they are essential for intracellular parasite viability [17,75,76].

The Tc24 antigen is another leading candidate for a vaccine against *T. cruzi*, since it is expressed in all developmental stages of *T. cruzi* and is a 24 kDa protein, which is encoded by multiple gene copies arranged in tandem arrays [77]. This antigen is located on the cell membrane and is primarily located at the flagellar pocket of *T. cruzi* and has calcium-binding domains [77,78]. Tc24 is a B-cell superantigen and possesses immunomodulatory properties, since Tc24 can be hydrolyzed by antibodies present in serum of unexposed mice and humans, and exposure to Tc24 eliminates the catalytic activity of the IgM molecules present in unexposed animals [79]. The administration of a DNA vaccine encoding Tc24 antigen can prevent Chagas disease progression both in murine and canine models [78]. Moreover, the administration of recombinant Tc24 protein can decrease parasitemia and cardiac parasite burden in immunized animals compared to controls [78,80]. Tc24 is highly conserved, especially the EF hand domains (structural domain helix-loop-helix found in calcium-binding proteins) and the calcium-binding loops, therefore Tc24 is a good vaccine antigen candidate [78,79].

New tools for the search of vaccine candidates exist in the post-genomic era. Genomic and proteomic information, along with bioinformatic tools are used to find new immunogens (reverse vaccinology) [81,82]. By analysing *T. cruzi* gene sequences, eight potential immunogens were identified by Bathia and colleagues, named TcG1-TcG8 [83]. DNA vaccines based on plasmids encoding TcG1, TcG2, and TcG4 antigens induced Th1 immune responses in mice that proved to be protective against a *T. cruzi* infection [31,83,84]. Those DNA vaccines were delivered with plasmids encoding IL-12 and Granulocyte-macrophage colony-stimulating factor (GM-CSF) as co-adjuvants. More recently, DNA prime and protein boost protocols based on TcG2 and TcG4 antigens were able to induce long lived anti-*T. cruzi* cell immunity [85].

Another new approach using antigens for vaccination comes from the fact that *T. cruzi* has a cell surface covered by immunogenic glycoconjugates. One of the immunodominant glycotypes, the trisaccharide Galα(1,3)Galβ(1,4)GlcNA, is expressed on GPI-anchored mucins of the infective trypomastigote stage of *T. cruzi* and triggers high levels of protective anti-α-Gal antibodies in infected individuals [86]. Vaccines based on this glycotype induce high antibody levels and protect against acute Chagas disease in a murine model [87].

### 2.2. Most Likely Candidate Antigens for Vaccine Development

Multiple antigen vaccines have been prepared with the aim of increasing the specificity and potency of the triggered immune responses. A multi-antigen approach, targeting several key gene products might result in better protection. One example of a trivalent vaccine is Traspain. The chimeric antigen includes the N-terminal domain of Cz, the central region of ASP2, and a subdominant region of inactive TS without the dominant TS epitope TSkb20, however, it contains a small amino acid region with alpha-helix structure [88]. Vaccination with this chimeric antigen and a new adjuvant triggered a strong immune response that proved to be directed against the immunogen and elicited both a B cell response and a Th1/Th17 response [88]. Another successful strategy of antigen combinations are ASP-2/TS [89], TcG2/TcG4 [90], a recombinant vaccine made of TS/Cz [91] and the trivalent Cz/Tc52/Tc24 [92]. Even though the major contribution to antibody response in mice was observed with Cz and Tc52, there was almost no contribution from the Tc24 antigen [93]. However, a vaccine with Tc24 as the unique antigen has proven to be highly protective in mice when tested during the chronic phase, since it protects from cardiac pathology and reduces parasitemia [94]. These vaccines can prevent cardiac fibrosis and necrosis, which are hallmarks of chronic Chagas disease cardiomyopathy. Interestingly a recent study has detected a recall memory immune response against recombinant TSA-1 and recombinant Tc24 antigens in mononuclear cells of asymptomatic chagasic chronic patients, suggesting that they are processed and induce a cellular immune response during natural infections [95]. These observations support the potential use of these antigens in humans [95]. All those antigens play an essential role in *T. cruzi* viability, are highly immunogenic, well conserved, located at the cell surface of the parasite and present in all developmental stages. Based on the above-described surface antigens in *T. cruzi*, we would suggest a chimeric trivalent antigen vaccine, contained in a single ORF in order to maximize efficacy. This putative ORF is described in Figure 2.

### 2.3. DNA-Based Vaccines

Genetically engineered DNA has been explored as a vaccination strategy over the last three decades. This strategy has been tested to immunize against several infectious diseases including HIV, hepatitis B, several parasites, viruses and also against cancer [96,97]. However, the success has been limited, mainly due to the low transfection efficacy and immunogenicity of the DNA-based vaccine. Perhaps one major breakthrough comes from the formulation and use of a modified chimpanzee DNA adenovirus vaccine against SARS-CoV-2 by AstraZeneca [98]. The DNA vector encodes the viral S-protein of SARS-CoV-2 and it is used as a template in human cells to produce multiple copies of the S-protein to generate an immune response against it [98]. Unlike conventional vaccines, instead of using an antigen from the pathogen, DNA-based vaccines carry a DNA vector encoding a specific antigen, which is expressed in the host to produce an immune response (see Figure 3). There are several benefits of DNA-based vaccines compared to conventional ones, such as they are safer for the personnel that produce the vaccine, cheaper, highly specific, and stable, and have the possibility of introducing additional genes for additional antigens or immunostimulatory molecules, such as cytokines. Moreover, some viral antigens need species-specific post-translational modifications, such as phosphorylation or glycosylation as the natural viral infection does, which are not present in recombinant antigens.

DNA-based vaccines are usually administered intramuscularly and lead to cell transfection at the injection site. Upon entry to the resident cells, skin keratinocytes and muscle cells, the DNA molecule is translocated to the nucleus, and is expressed and secreted by exosomes or apoptosis [97]. Antigen presenting cells, such as macrophages and dendritic cells, which circulate looking for foreign pathogen structures, are able to internalize the antigens and migrate to the lymph nodes to present peptide fragments (bound to MHC molecules) to immune cells to produce a B cell (humoral) and T cell (cytotoxic) immune response. Additionally, DNA vaccines can directly transfect macrophages and dendritic cells to express the antigen and be presented to immune system cells [97].

Typically, a standard DNA vector for DNA-based vaccines contains an expression unit (or transcription) and a production unit (Figure 4). The expression unit consists of a viral-hybrid or strong eukaryotic promoter (region I), an intron (region II), the antigen coding sequence (region III) and a polyadenylation signal (region IV, poly A signal) [97]. The promoter provides a strong RNA polymerase II binding site to transcribe the mRNA and the intron, which is introduced between region I and III and can increase the antigen expression significantly [97]. The viral promoter becomes quickly inactivated by gene silencing and they often show only a transient expression [97,99] and we suggest the use of the strong super core promoters designed by Kadonaga and colleagues to avoid gene silencing and obtain strong gene expression [100]. The poly A signal (region IV) stabilizes the mRNA, facilitates the export from the nucleus and helps the protein translation process. On the other hand, the production unit contains a bacterial origin of replication (region V), and a selectable marker (region VI) to be produced in bacteria [97]. However, the legislation strongly discourages the use of antibiotic resistance genes as a selectable marker for clinical uses and also the use of any sequence that could promote or produce an integration event into the host DNA that might lead to mutations [97].

As for DNA-based vaccines against *T. cruzi*, similar considerations for RNA vaccines must be taken into account to select an antigenic molecule from the parasite cell’s surface. Consequently, a highly constant antigenic region must be chosen by means of antigenic programs, which can be back translated according to human codon usage and cloned into region III. Additionally, either a chimeric molecule (from two or more different proteins) and a coding region for an immunostimulatory molecule can be introduced. Vaccines should be delivered into the appropriate co-adjuvant to enhance stability, cellular uptake, specific targeting, and immunomodulation [97]. DNA vaccines are usually delivered into liposomes, although other carriers can be also used. A typical liposome to deliver DNA vaccines is described in Figure 5. Co-adjuvants and carriers for DNA-based vaccines are discussed in the next sections of the manuscript.

### 2.4. RNA-Based Vaccines

Conventional vaccination is one of the main breakthroughs in modern medicine, which has reduced the incidence of deadly infectious diseases. However, this strategy has not been effective against other emerging infectious diseases, such as those caused by HIV, Zika or Ebola viruses. Therefore, messenger RNA (mRNA)-based vaccines are a good alternative to fight those infectious diseases [101,102,103]. Nowadays, we are facing a dangerous pandemic caused by the SARS-CoV-2 virus, which causes the infectious COVID-19 disease that has caused more than 434 million infected people and 5.9 million deaths (World Health Organization, https://covid19.who.int/ last accessed on 1 February 2022). Moreover, this pandemic disease is devastating the economies of most of the countries around the world. Vaccines against the SARS-CoV-2 virus have been produced that are helping to reduce the number of infected people and deaths and it is expected that soon this pandemic disease can be controlled. Vaccines against this virus have been formulated based on conventional attenuated viruses, DNA-based vaccines, virus recombinant proteins and RNA-based vaccines [104]. Two of the leading vaccines have been formulated based on messenger RNA (mRNA) of the Spike S protein (S) of the virus surface, which binds to the angiotensin-converting 2 enzyme (ACE2) receptor that is used by the virus to enter and infect host cells [104]. Those two vaccines have been developed, manufactured, and sold by Moderna (Cambridge, MA, USA) and Pfizer (Berlin, Germany) and have been proven to be one of the most effective vaccines against the virus to prevent new infections and death of infected patients [104]. Vaccines based on mRNA could have several advantages, since they are faster and cheaper to produce, safer for both the personnel that produce it and the patient due to the fact that they do not use highly infectious viruses. Additionally, mRNA vaccines do not possess oncogenic potential via integration into the host DNA, they only need to reach the cytoplasm to be translated by the ribosome and additionally mRNA is more immunogenic than DNA. Taking those advantages altogether, we suggest a strategy to formulate mRNA-based vaccines against *T. cruzi* to control this infectious disease.

A high awareness of mRNA biology is necessary to design a mRNA-based vaccine. A eukaryotic mRNA is composed of a coding region (open reading frame, ORF), which is flanked by 5′ and 3′ untranslated regions (UTRs), a 5′ 7-methylguanosine triphosphate (m7G, 5′ cap) and a 3′ poly (A) tail, which enhance mRNA stability and the protein translation process [105,106,107,108]. Though, the poly (A) tail and the 5′ and 3′ UTRs can be included in the construct design, the m7G should be introduced into the mRNA by using 5′ cap analogues or by using vaccine capping enzyme, a process that is not 100% efficient, therefore a portion of the mRNA is not capped. The resulting uncapped mRNA is not a good template for protein synthesis and thus the amount of mRNA for translation is much less [105,106,107,108]. DNA templates used to produce mRNA typically contain a bacterial promoter for the RNA polymerase to start transcription at the initiation site, the mRNA to be transcribed, with the above-described elements, and elements to be amplified in bacterial systems, such as a bacterial replication origin and a selectable marker [108]. The mRNA is in vitro transcribed from the plasmid DNA by using a bacterial RNA polymerase and then 5′ capped. The construct is described in Figure 6 and the steps to design and produce a functional mRNA for vaccination is described in Figure 7.

As a first part of the strategy, an antigen from the parasite surface, which must be highly constant and less prone to mutations can be selected using antigenic prediction programs, then this antigenic region should be back translated to the optimal host codon usage to be cloned into a vector, in vitro transcribed (IVT) and 5′-capped to serve as a template for protein translation and then injected into the host with an appropriate co-adjuvant (see section below). The mRNA will be delivered to myocytes, which can translate this template into a polypeptide, secrete it, and the protein antigen will be taken up by macrophages and dendritic cells to present the antigen to immune system cells to mount an adaptive B and T cell response. The whole strategy is described in Figure 7.

Nowadays, the potential of mRNA has been increasingly recognized in regenerative medicine, immunotherapy, vaccination, and gene editing, however, to fully potentiate its applications efficient production, stabilization and delivery strategies into the target cells are necessary. The therapeutic use of mRNA is undoubtedly recognized, and it has been greatly reinforced by the mRNA-based vaccines against the SARS-CoV-2 virus (causing the COVID-19 disease), developed mainly by Pfizer and Moderna to fight the pandemic.

### 2.5. Live Attenuated Vaccines

Presently, a lot of efforts have been made to develop a new generation of live attenuated vaccines (LAV) to provide long-term immunity against protozoan diseases, including leishmaniasis, malaria and Chagas disease [109,110]. In early studies of *T. cruzi* infection, mainly in mouse models, it has been shown that mice which survived an acute infection were resistant to reinfection [110]. This immunity relies on a parasite specific Th1 response together with an antibody response against the parasite [110]. Also, cytotoxic CD8+ T cells were produced, which are key in the development of an effective immune response, which is maintained during the chronic phase of the infection [111], however, a sustained cytotoxic response during the chronic phase would lead to myocardial damage. Therefore, it is necessary to produce a vaccine which can rapidly control the acute infection and be able to down modulate the aberrant immune response associated with the parasite during the chronic stage of the infection.

One of the first efforts to develop a vaccine against Chagas disease was immunization with a *T. cruzi* strain (TCC), attenuated by culture passage [109,110]. This vaccine proved to be safe, and it can control parasitemia after a subsequent challenge with trypomastigotes from the highly virulent Tulahuen strain. This vaccine can reduce the transmissibility and myocardial tissue damage in mice and dogs [112,113]. *T. cruzi* live attenuated strains can be produced by irradiation, heat treatment, formalin treatment, culture passage and genetic manipulation, including CRISPR technology [109,110]. Genetic manipulation has allowed the production of several genetically attenuated parasite cell lines (GAPs), which have been used as vaccines and tested in mice [109,110]. One such example is the deletion of an allele of the calmodulin-ubiquitin gene in the Tulahuen strain, making the attenuated vaccine TulCub8, which can reduce parasite load after infection with the wild type strain in mice [114]. Another example is the attenuated L16 line, in which the lyt-1 (encoding for a virulence factor) gene has been deleted, which can confer immunity to parasitemia for at least 14 months after vaccination in mice [115]. All of these observations make the generation of GAPs that could be tested in humans against Chagas disease attractive. However, the risk of reversion of the GAP lines to a virulent phenotype and the idea that cardiopathy in the chronic phase might be related to parasite presence, could limit the use of those vaccines.

### 2.6. T4 Bacteriophage Nanoparticles for Vaccine Delivery

Subunit vaccines, which can contain one or more target antigens from the pathogen, are safer than whole pathogen vaccines. However, the target antigens are not immunogenic and require adjuvants to be immunogenic and to provide protection. Therefore, new approaches should be used to deliver subunit vaccines in an immunogenic fashion. The assembly of antigens into virus-like particles (VLPs) could be a better approach for subunit vaccine delivery against infectious diseases [116,117]. VLPs might better present the antigens and stimulate the host innate and adaptive immune response to elicit both humoral and cellular immune responses [116]. Phage T4 is a bacteriophage that can provide an excellent platform to generate the nanoparticle subunit vaccines. T4 phage has three major components namely head (or capsid), tail and tail fibers [118]. The application of T4 phage in VLP vaccine development mainly involves the head, since it contains two non-essential outer proteins called Soc and Hoc, in which the antigens are fused [116,117,118]. Soc and Hoc allow a high-density array of antigen epitopes in the form of domains, peptides, full length polypeptides or even multi-subunit complexes. The antigens in the VLPs show a repetitive, symmetrical, and high-density array, which resembles the pathogen-associated molecular patterns that are present in bacteria and viruses [116]. Those antigens are highly immunogenic and do not need adjuvants to elicit an immune response, which can protect against bacterial and viral pathogens. Presently, VLP vaccines based on T4 phage have been developed against viruses and bacteria and tested in animal models, mainly in the mouse model [117]. VLP vaccines have been described for Bacillus anthracis, Yersinia pestis, foot-and-mouth disease virus, classical swine fever virus and bursal disease virus [117]. However, VLP vaccines have not been developed yet for protozoan diseases, but it is tempting to speculate that VLP vaccines could be developed for Chagas disease. Those subunit VLP vaccines could be monovalent or polyvalent and they might be an alternative to other vaccine development approaches.

### 2.7. Co-Adjuvants

#### 2.7.1. Co-Adjuvants for Protein Antigen-Based Vaccines

The role of vaccine adjuvants has been crucial to determining the immunogenicity of the antigen and orchestrate an adequate adaptive host immune response to counteract *T. cruzi* infection. Bacterial, viral, and parasitic DNA are quite different from mammals, and they are rich in CpG oligo-deoxy nucleotide sequences, which are immunostimulatory motifs and are easily detected by the Toll-like receptor 9 [119]. This allows the induction of a Th1 immune response in subunit vaccines against intracellular pathogens by using CpG oligo-deoxy nucleotides as co-adjuvants [120,121,122]. The elicited Th1 response with *T. cruzi* Cz and CpG was characterized by a strong antibody response of the IgG2a class, and is quite different to the same antigen with alum as adjuvant, which displayed an IgG1 response instead [123]. Moreover, splenocytes of the Cz-CpG group showed strong proliferation with high levels of IL-2 and IFN-γ secretion, a Th1 response and a much better protection when the immunized animals were challenged with *T. cruzi* trypomastigotes [122]. Also, the capacity of Cz as an antigen candidate for a Chagas disease vaccine was determined by the combined use with IL-12 and a neutralizing IL-4 monoclonal antibody that conferred protection to mice challenged with *T. cruzi* trypomastigotes [123,124]. The Th1-oriented immune response induced by using CpG oligo-deoxy nucleotides as co-adjuvant has been demonstrated to be protective not only when using Cz as the antigen. The amastigote surface protein 2 (ASP-2) together with CpG oligo-deoxy nucleotides, provided immunity with 100% protection to infection with *T. cruzi* in mice [125]. Another similar case is with TS, when combined with CpG oligo-deoxy nucleotides, which provides mucosal and systemic immunity against *T. cruzi* infection [126,127]. Finally, Tc52 and Tc80 when combined with CpG oligo-deoxy nucleotides also protects against *T. cruzi* infection [59,128]. The immunization with CpG oligo-deoxy nucleotides, to stimulate IFN-γ synthesis, also induced IL-10 secretion in a strongly Th1-oriented immune response [31,123]. Considering that IL-10 produces an anti-inflammatory response, this regulatory component would prevent a severe *T. cruzi* infection outcome.

Another vaccine adjuvant was tested in combination with Cz, the synthetic derivative of the macrophage-activating lipopeptide *Mycoplasma fermentans* (MALP-2), an adjuvant which improves the humoral and cell mediated immunity by activation of Toll-like receptor 2/6 (TLR2/6) [129]. The immunization of Cz plus MALP-2 proved to be effective in controlling a challenge with trypomastigotes [129]. The elicited immune response with recombinant Cz induces IgG1 specific antibodies that can be switched to IgG2a when Cz is combined with CpG oligo-deoxy nucleotides. The switch can be improved by an intranasal boost of Cz plus MALP-2, that also switches the immune response to Th1, as demonstrated by the release of IFN-γ by the high number of IFN-γ -producing T cells [129].

Last generation co-adjuvants that induce new immunological mechanisms have been used and they are able to confer immune protection. Vaccines with Tc52 and cyclic dinucleotides such as cyclic diAMP (CDA) have been shown to induce protective immune responses against *T. cruzi* infection with a mixed Th1/Th17 profile [60]. The CDA-adjuvant vaccine induced more IL-17 secretion than the vaccine based on CpG oligo-deoxy nucleotides. The IL-17 stimulation was correlated with the protective ability of the vaccine, since the addition of a derivative of the alpha-galactosyl-ceramide, a Th17 inhibitor, decreased the vaccine efficacy [31,130].

#### 2.7.2. Co-Adjuvants for Nucleic Acid-Based Vaccines

The above-described co-adjuvants can be used with protein antigens; however, a different co-adjuvant series must be used for DNA and RNA-based vaccines. Nucleic acids can be delivered to the cytoplasm of eukaryotic cells by several strategies including naked nucleic acids, gene gun delivery methods, protamine condensation, adjuvants, nanoparticles of cationic liposomes and biopolymers [97,131,132,133,134]. Those strategies are intended to provide stability to the nucleic acid and to enhance the immune response. We will describe only those strategies based on nanoparticles of cationic liposomes and biopolymers.

Biopolymers to deliver DNA-based vaccines can be defined as polymers made by living organisms such as corn, seaweed, and crustaceans and they show low toxicity, biocompatibility, great physicochemical versability, favorable cellular interactions, biodegradability, and easy production [97]. The DNA can be adsorbed, incorporated inside the biopolymer matrix, or encapsulated into the biopolymer matrix [97]. The DNA-encoding antigen should reach the target cell, internalized into the cell by phagocytosis or endocytosis, escape from the phagosome or endosome vesicles, reach the nucleus, and dissociate from the carrier, to be transcribed and the mRNA exported to the cytoplasm for translation. The DNA carrier for vaccines should ensure stability of the DNA molecule, possess functional chemical groups to provide stability in extra and intracellular fluids, overcome the extracellular and intracellular barriers, provide a prolonged circulation to reach the target cells, should stimulate the immune response as long as possible and ensure safe delivery to the nucleus. A more detailed description of biopolymeric carriers is presented in reference [97].

Liposomes are another type of carrier to deliver DNA-based vaccines. Liposomes should provide stability to the DNA, protect it against nuclease attack, promote cell transfection efficiency by enhancing cellular uptake and provide safe delivery into the nucleus. The liposomes consist of cationic lipids forming one or several lipid bilayers with the DNA encapsulated within it. Positively charged cationic liposomes are useful carriers for negatively charged DNA [134]. Moreover, they are also suitable to carry co-adjuvants such as CpG oligodeoxynucleotides to enhance the immune response. A DNA liposome vaccine is represented in Figure 5. To design an efficient vaccine against *T. cruzi*, those properties described in Figure 5 should be considered.

RNA-based vaccines are delivered together with molecules such as poly I:C RNA or CpG oligo deoxy-nucleotides, to enhance immunogenicity and stability, similar to protein antigen-based vaccines [135,136]. However, cationic liposomes have been successfully used to deliver mRNA-based vaccines. These consist of closed lipid bilayer vesicles, which can be formed spontaneously in water, and consist of one or several lipid bilayers, which can encapsulate the mRNA [135,136]. They are useful for delivering a variety of nanomedicines, such as proteins, enzymes, and drugs. Hydrophilic molecules can be encapsulated into the aqueous interior of the liposome, while hydrophobic molecules can be entrapped in the hydrocarbon chain region of the lipid bilayer. Phospholipids, such as phosphatidylcholines, phosphatidylethanolamines, phosphatidyl serines, and phosphatidylglycerol are stabilized by cholesterol which are common liposome constituents [135]. Presently, lipid nanoparticles have emerged as promising tools to deliver mRNA-based vaccines and a variety of therapeutic agents. Currently, cationic lipid nanoparticles are in the spotlight as components of the mRNA-based vaccine against SARS-CoV-2 virus. Cationic lipid nanoparticles exhibit a more complex architecture than liposomes and enhanced physical stability. A comprehensive review on cationic lipid nanoparticles can be found in reference [135] and the components used to make the cationic lipid nanoparticles used in the mRNA-based vaccine against SARS-CoV-2 can be found in references [104,135,136].

For Chagas disease, DNA-based vaccines have been developed and delivered with different co-adjuvants [137,138]. DNA vaccines can provide an alternative for both prevention and treatment of a variety of infectious diseases, including Chagas disease. We suggest that suitable DNA vectors encoding antigens from *T. cruzi* could be encapsulated into cationic lipid nanoparticles and used as vaccine systems to deliver and obtain an efficient vaccine against *T. cruzi*.

Lastly, several nucleic acid vaccine carriers have been successfully used to deliver vaccines. These include liposomes, polymers, biopolymers, virosomes, cell-penetrating peptides (CPPs) and live bacteria. The nucleic acid vaccine carriers are shown in Figure 8.

## 3. Conclusions

Nucleic acid (RNA and DNA)-based vaccines have been developed as an alternative to conventional vaccines. Until the past year, those vaccines were mostly in clinical trial phases, however, this present year due to the SARS-CoV-2 pandemic they were quickly developed and used successfully to immunize against the virus that causes the COVID-19 disease. It is expected that their success will boost the development of another nucleic acid-based vaccine, especially against those infectious diseases where the conventional vaccines have failed or against deadly diseases that can be prevented by means of vaccination such as cancer.

Chagas disease is one of the infectious diseases in which despite all investigation and efforts, does not have a successful pharmacological treatment or vaccine to effectively treat the disease. Therefore, we suggest a strategy to design and develop a vaccine based on this new technology, which has a series of advantages compared with conventional vaccines. Nucleic acid-based vaccines are cheaper, easier to make, safer for the personnel that produce it, versatile, and easier to formulate than conventional ones. They offer the possibility of the use of chimeric antigens, polyvalent antigens and immunostimulatory molecules that can be incorporated in the transcriptional units. However, there are still several issues to be resolved such as the use of genetic material in a safe way and the low immunogenicity of the first nucleic acid vaccine formulations. Some of those obstacles can be resolved by using DNA/RNA nanocarriers composed of lipids, biopolymers, and inorganic compounds.

In the next few years, we will see several nucleic acid vaccines going to clinicals trials and approved for use against infectious diseases or against cancer and most likely combinatorial therapies will be used. Such is the case with Chagas diseases, where anti-inflammatory and antioxidant agents together with vaccines can manage the chronic complications of this disease.

## Figures and Tables

**Figure 1 vaccines-10-00587-f001:**
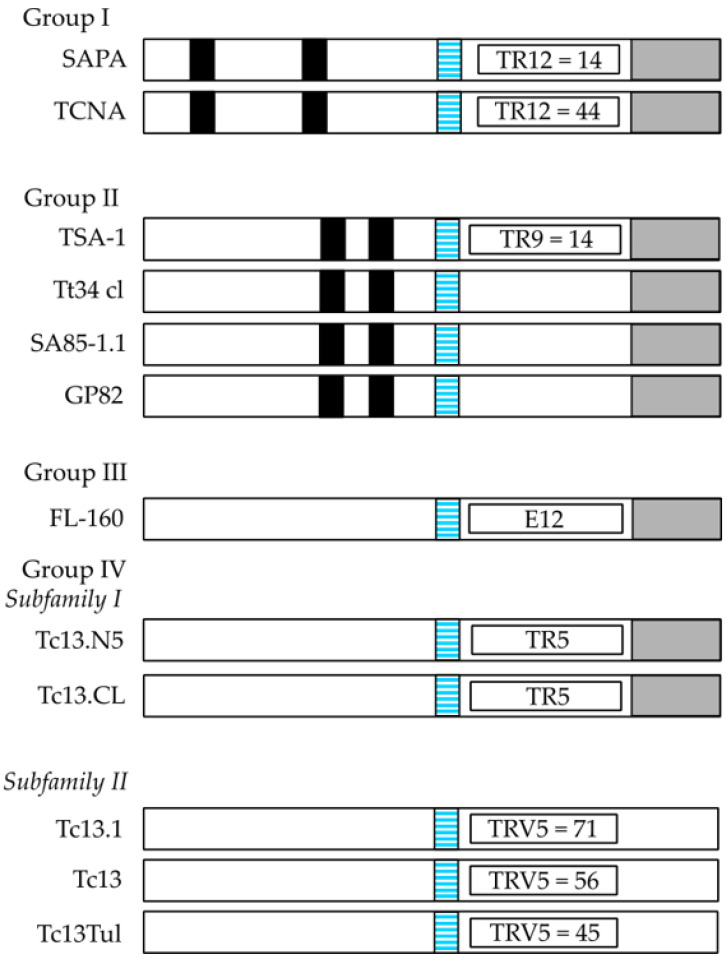
The Trans-sialidase (TS) superfamily. Schematic representation (not to scale) of the four groups of TSs from *T. cruzi*. Characteristic motifs for TS are SxDxGTW (black boxes), VTVxNVxLYNR (light blue boxes), and the GPI-anchor signal at the C-terminus is shown in grey. Tandem repeats (TR) containing 12 amino acid residues [DS_2_AH(S/G)TPSTP(A/V)] are shown inside an open box (TR12) and they are detected in SAPA and TCNA. In TSA-1, nine amino acid residue repeats (DK_2_ESESGDSE) are present (TR9, open box). This repeat is present 14 times in TSA-1. FL-160 possesses a characteristic epitope TPQRKT_2_EDRPQ (E12, open box). In subfamily I of Group IV, a pentapeptide (EPKSA) is found once (TR5, open box), whereas in subfamily II it is repeatedly found (TRV5, open box). In members of subfamily II of Group IV a GPI-anchor signal is lacking.

**Figure 2 vaccines-10-00587-f002:**
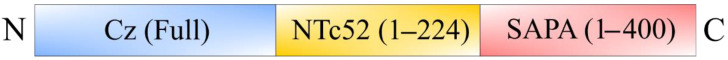
A chimeric antigen for vaccination against Chagas disease. A putative trivalent vaccine including the antigenic regions of full Cz, N-terminal Tc52 (NTc52 from 1–224) and SAPA from 1–400. The cDNA encoding that highly immunogenic fusion polypeptide can be synthesized and used either for a mRNA-based vaccine or a DNA-based vaccine against Chagas disease.

**Figure 3 vaccines-10-00587-f003:**
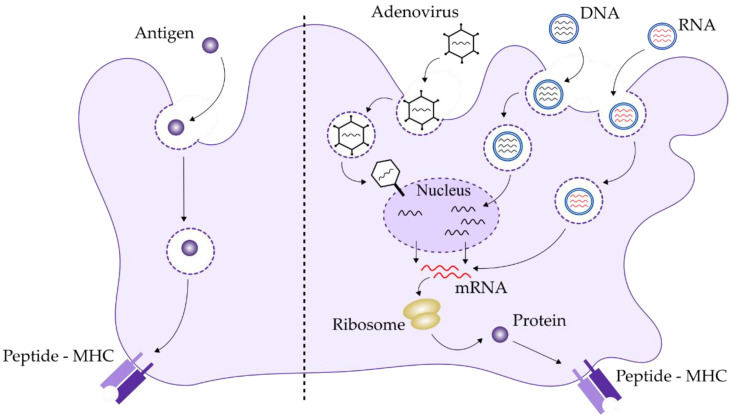
Schematic representation of the differences between conventional and nucleic acid-based vaccines. Conventional vaccines include inactivated pathogens, pathogen subunits or live-attenuated pathogens which are taken up by antigen presenting cells and presented to the immune system cells (on the left). On the other hand, nucleic acid-based vaccines (on the right) are delivered using adenovirus or liposomes which are able to fuse to the plasma membrane and released into the cytosol where the DNA vaccines enter the nucleus, and are transcribed and the mRNA is translated inside the ribosomes and the protein is presented to the immune system cells. RNA vaccines are released into the cytosol and translated inside the ribosomes and the polypeptide is presented to the immune system cells.

**Figure 4 vaccines-10-00587-f004:**
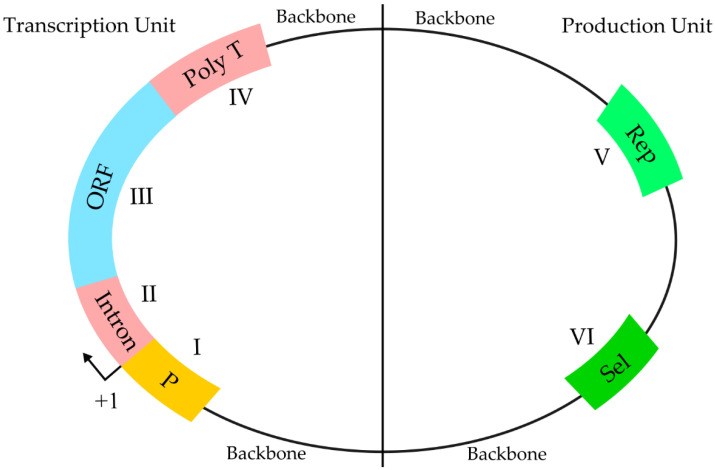
Schematic design of a plasmid DNA to be used in DNA vaccines. The transcription unit (on the left) contains several regions which includes a eukaryotic RNA polymerase II promoter (I), an intron (II) an antigen coding sequence (III) and a poly T region (IV) to produce a poly A tail in the transcript which can ease the export from the nucleus, stabilize the transcript and help in the translation process. The production unit (on the right) consists of a replication origin (V) for successful replication in the bacterial host and an antibiotic selection marker (VI).

**Figure 5 vaccines-10-00587-f005:**
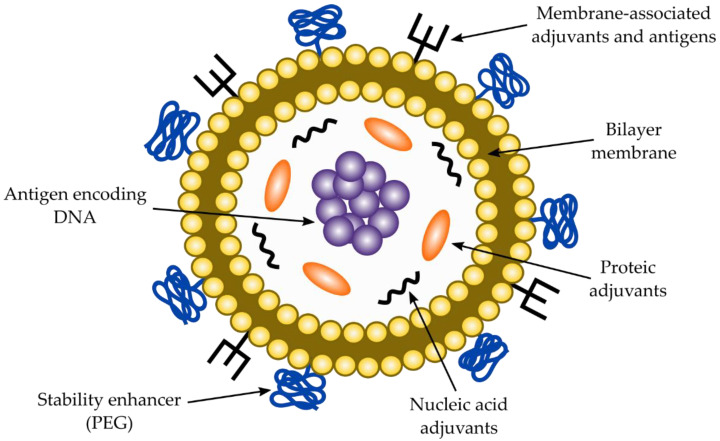
Schematic representation of a DNA liposome vaccine. The incorporation of the antigen-encoding DNA can be either in the aqueous inner space or by integration into the bilayer membrane, which will depend on the lipophilic properties of the compounds. Additionally, the liposome might carry PEG (stability), nucleic acid adjuvants (poly I:C, CpG), protein adjuvants (cytokines, immunostimulant molecules), membrane associated antigens and adjuvants (glycolipids, lipopeptides, cationic adjuvants) and the lipid bilayer membrane (cationic, anionic, or neutral). Note that mRNA can be also used to incorporate into liposomes to produce mRNA vaccines.

**Figure 6 vaccines-10-00587-f006:**
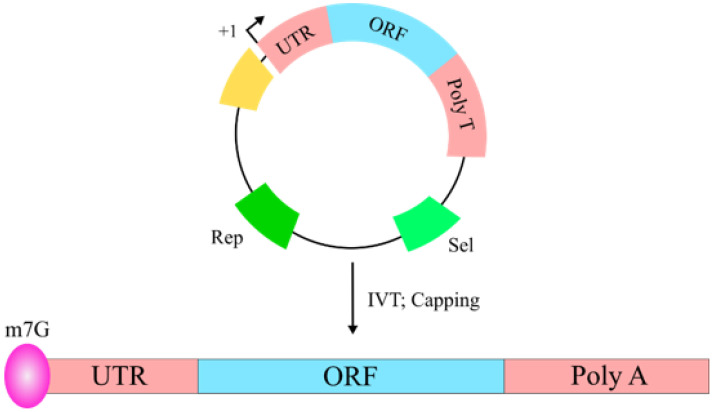
Representation of a vector to produce mRNA-based vaccines. The vector contains the basic elements to produce a functional mRNA to be used in vaccination. This vector contains a strong viral RNA polymerase promoter (yellow), an initiation site (+1), an untranslated 3′ sequence (UTR, enhances translation), the ORF which encodes the antigen, a poly T tail (produce a poly A to enhance translation), a selection marker (Sel) and an origin of replication (Rep). After IVT, the mRNA is capped to produce a m7G mRNA, which can be purified and used for vaccination.

**Figure 7 vaccines-10-00587-f007:**
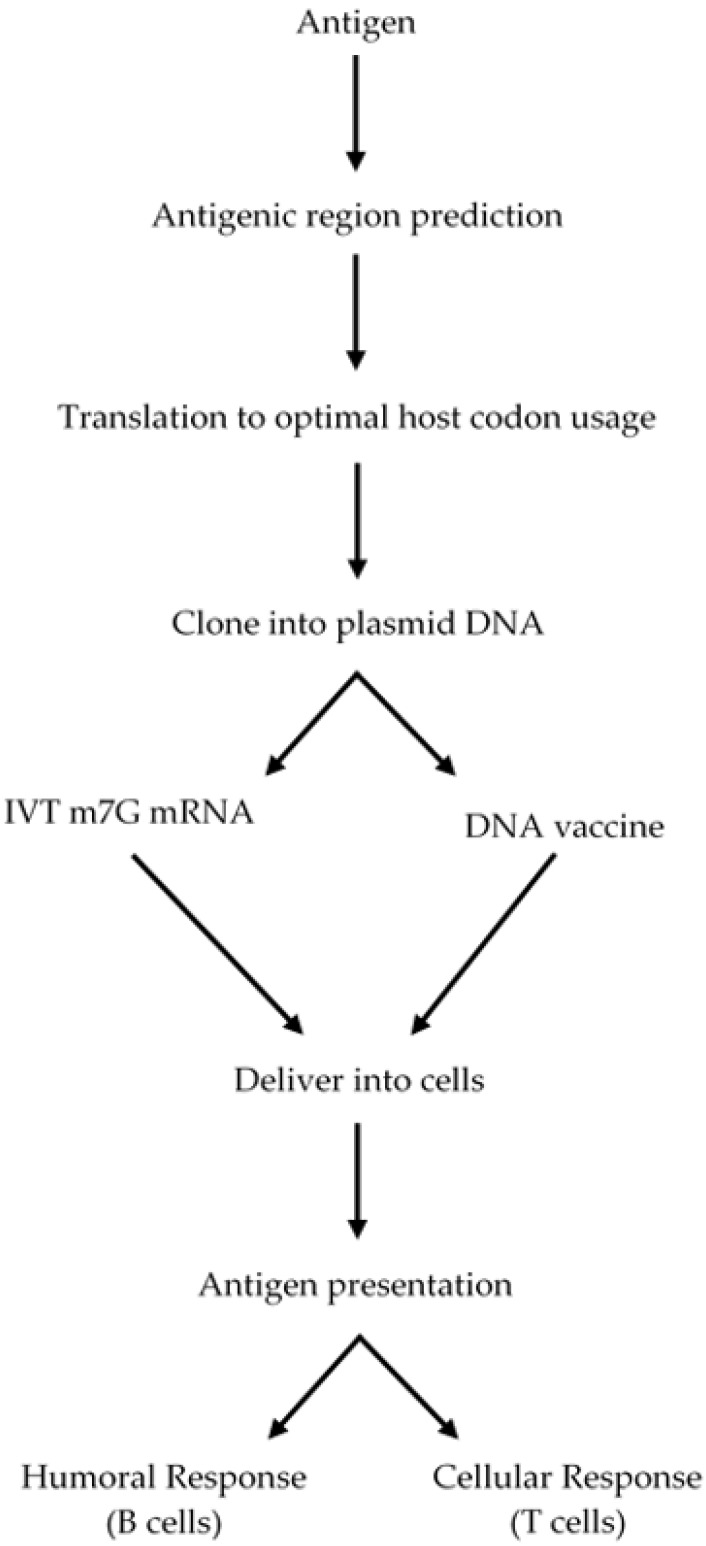
A strategy to design a mRNA-based vaccine. A polypeptide (antigen) is chosen to develop a vaccine and then the whole polypeptide or antigenic regions are selected by using bioinformatic programs. The cDNA is synthesized using the codon usage of the species to be vaccinated. The cDNA is cloned into a plasmid DNA vector as described in Figure 6 and then in vitro transcribed (IVT) and capped (m7G mRNA). The m7G mRNA is purified and incorporated into liposomes or another carrier and then injected into the individuals. The m7G mRNA enters the cytosol and is translated by the ribosomes and presented to immune system cells (Figure 3) to produce antibodies, cytotoxic T cells and B and T memory cells.

**Figure 8 vaccines-10-00587-f008:**
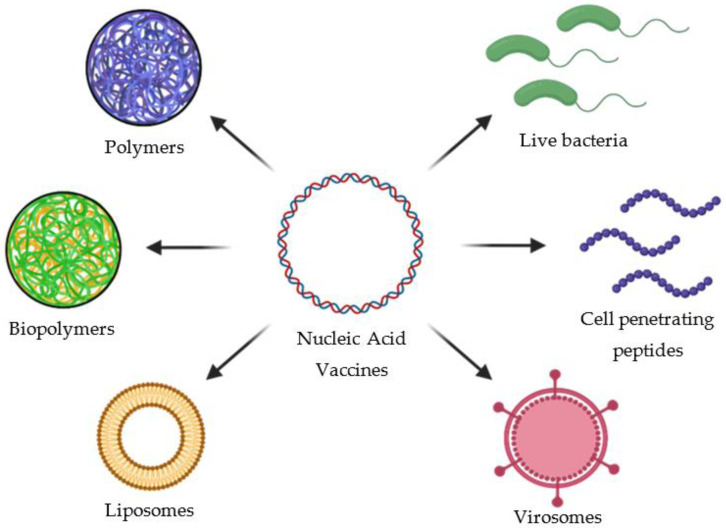
Nucleic acid vaccine carriers. In the figure is shown the most used carriers to deliver nucleic acid-based vaccines such as liposomes, virosomes, polymers, biopolymers, cell penetrating peptides (CPPs) and live bacteria.

## Data Availability

All data are included within the manuscript.

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
