# Peer review of "Vaccine Design against Chagas Disease Focused on the Use of Nucleic Acids"

_vaccines, 2022, doi:10.3390/vaccines10040587_

Round 1

Reviewer 1 Report

The work by E. Maldonado, A. Solari et al. is a review about the progress in the development of vaccines for Chagas disease. In the review they clearly expose the work made by others classified into the types of vaccines specifying each particular antigen to target. I would recommend its publication with previous corrections of a few errors I found during the reading listed below:

-in the abstract second row says "affcets" and should say "affects".

-in the abstract seventh row says "wich" and should say "which".

-names of compounds should not be capitalized like "Benznidazole and Nifurtimox " in the second page.

-in page 3 says "a N-terminal " but should say "an N-terminal"

Author Response

Reviewer 1.

All those spelling mistakes were revised and corrected.

Reviewer 2 Report

In this Review, authors describe the design of vaccines against Chagas disease, focused on the use of nucleic acids. The review is interesting and relevant in the presented form; however, some points should be considered for publication in Vaccines.

1) Considering that it is a review about vaccines against Chagas disease, it would be interesting to include data related to live attenuated vaccines against T. cruzi. This strategy has been tested using different attenuated lines with relative protection in animals models. This strategy has been considered as the most promising for vaccines against intracellular parasites, particularly for T. cruzi and Leishmania. [Check: https://pubmed.ncbi.nlm.nih.gov/34896016/]. What are the advantages and disadvantages compared to vaccines based on nucleic acids? I think these points should be included in this review.

2) Page 2 – “v) the candidate molecule should not undergo mutations.”What do the authors want to know here? Please specify.

3) Page 8 – On item “2.3. DNA-based vaccines”, the authors state that Aztrazeneca vaccine against SARS-CoV-2 is a DNA based vaccine. This is not correct. In fact, this vaccine is a chimp adenovirus that contains the coding sequence for the spike protein of SARS-CoV-2. This point must be corrected in the review. Though, figure 3 must also be revised once that Aztrazeneca vaccine is not a nucleic acid-based vaccine that is delivered in liposomes.

Author Response

Reviewer 2

  1. We added a section “2.5. Live attenuated vaccines” on live attenuated vaccines and their advantages and main concerns about their use.
  2. We meant “antigenic variation by a high rate of mutations”
  3. We are sorry for that mistake, and it has been corrected according.